# Complex Changes in the Efficiency of the Expression of Many Genes in Monogenic Diseases, Mucopolysaccharidoses, May Arise from Significant Disturbances in the Levels of Factors Involved in the Gene Expression Regulation Processes

**DOI:** 10.3390/genes13040593

**Published:** 2022-03-26

**Authors:** Zuzanna Cyske, Lidia Gaffke, Karolina Pierzynowska, Grzegorz Węgrzyn

**Affiliations:** Department of Molecular Biology, Faculty of Biology, University of Gdansk, Wita Stwosza 59, 80-308 Gdansk, Poland; zuzanna.cyske@phdstud.ug.edu.pl (Z.C.); lidia.gaffke@ug.edu.pl (L.G.); karolina.pierzynowska@ug.edu.pl (K.P.)

**Keywords:** mucopolysaccharidosis, gene expression, glycosaminoglycans, transcriptomics

## Abstract

Monogenic diseases are primarily caused by mutations in a single gene; thus, they are commonly recognized as genetic disorders with the simplest mechanisms. However, recent studies have indicated that the molecular mechanisms of monogenic diseases can be unexpectedly complicated, and their understanding requires complex studies at the molecular level. Previously, we have demonstrated that in mucopolysaccharidoses (MPS), a group of monogenic lysosomal storage diseases, several hundreds of genes reveal significant changes in the expression of various genes. Although the secondary effects of the primary biochemical defect and the inefficient degradation of glycosaminoglycans (GAGs) might be considered, the scale of the changes in the expression of a large fraction of genes cannot be explained by a block in one biochemical pathway. Here, we demonstrate that in cellular models of 11 types of MPS, the expression of genes coding for proteins involved in the regulation of the expression of many other genes at various stages (such as signal transduction, transcription, splicing, RNA degradation, translation, and others) is significantly disturbed relative to the control cells. This conclusion was based on transcriptomic studies, supported by biochemical analyses of levels of selected proteins encoded by genes revealing an especially high level of dysregulation in MPS (EXOSC9, SRSF10, RPL23, and NOTCH3 proteins were investigated). Interestingly, the reduction in GAGs levels, through the inhibition of their synthesis normalized the amounts of EXOSC9, RPL23, and NOTCH3 in some (but not all) MPS types, while the levels of SRSF10 could not be corrected in this way. These results indicate that different mechanisms are involved in the dysregulation of the expression of various genes in MPS, pointing to a potential explanation for the inability of some therapies (such as enzyme replacement therapy or substrate reduction therapy) to fully correct the physiology of MPS patients. We suggest that the disturbed expression of some genes, which appears as secondary or tertiary effects of GAG storage, might not be reversible, even after a reduction in the amounts of the storage material.

## 1. Introduction

Among about 20,000 known diseases caused by genetic defects, over 7000 are monogenic disorders, defined as diseases resulting from mutations in a single gene [1]. It is estimated that 1 in 50 humans suffer from such a disease [2]. Since the dysfunction of a single gene should result in the disturbance of the functions of just one protein or functional RNA species, one might predict that the mechanisms of monogenic diseases should be relatively simple. Following this path of thinking, the correction of this single gene or its product should result in the normalization of all functions of the affected organism. However, such predictions have turned out to be incorrect. It is not only the correction of genetic defects that have appeared to be extremely difficult at the organismal level, but also the effects of single mutations are significantly more complex than the disruption of one biochemical reaction in the cell. At the organismal level, it has been established that the same mutation can have different specific effects in different individuals, as indicated by various clinical symptoms [3]. Moreover, it has been demonstrated that mutations proven to cause pathogenic effects in the vast majority of patients may be asymptomatic in some persons [4]. The reasons for such huge variability in the effects of the same mutation remain unclear; therefore, studies on the details of the molecular mechanisms of monogenic diseases have been substantiated. The results of these investigations not only indicate the crucial regulations of biological processes, but they can also provide the bases for the development of novel, more effective therapeutical approaches.

Our recent studies on mucopolysaccharidoses (MPS), a group of monogenic inherited metabolic diseases, demonstrated an unexpectedly high number (between about ~300 and ~900, depending on the MPS type) of genes whose expression was significantly dysregulated in cells derived from patients relative to control cells [5]. MPS are rare diseases caused by mutations in the genes coding for lysosomal enzymes involved in the degradation of glycosaminoglycans (GAGs) [6]. Thus, a lack or drastic decrease in the activity of one such enzyme results in severely impaired GAG degradation and the accumulation of these complex carbohydrates in lysosomes, and further in the cytoplasm and outside the cells [7]. Depending on the kind of deficient enzyme and type(s) of accumulated GAG(s), different MPS types can be distinguished [8]. Nevertheless, all these types are severe diseases, with dysfunctions of most, if not all, organs and systems which lead to a considerable shortening of the life span, estimated to be 1–2 decades on average [9]. Current therapeutic options for MPS include bone marrow or hematopoietic stem cell transplantation (HSCT) and enzyme replacement therapy (ERT), and substrate reduction therapy (SRT), pharmaceutical chaperones, and gene therapy are under development [8,9]. However, none of these existing and emerging therapies can correct all the symptoms of MPS which might appear unexpected, as ERT provides the otherwise lacking enzyme and SRT restores the balance between GAG synthesis and degradation. In fact, the failure to cure MPS patients is not only due to problems with crossing the blood–brain barrier by the drugs, but also other, as yet poorly defined problems with the efficacy [10,11,12,13,14,15,16]. In this light, we proposed a hypothesis that secondary and tertiary changes in various cellular processes might not be reversible even if GAG levels are normalized [17]. The question remained: what are mechanisms of these disruptions in cell physiology? The discovery of massive dysregulations in the expression of hundreds of genes, as mentioned above [5], might provide a plausible explanation, while the scale of this phenomenon still appears astonishing if no general reason is found.

Among hundreds of genes with considerably changed levels of transcripts in MPS cells [5], there were those coding for proteins involved in different biological processes such as cell activation [18], regulation of cellular processes [19], proteasome composition and activity [20], apoptosis [21], morphologies of organelles [22], ion homeostasis [23], and even susceptibility to COVID-19 [24] and behavioral disorders [25]. This suggests that a more general mechanism might exist that could lead to high levels of differential dysregulation of gene expression control processes. Therefore, in this study, we investigated the expression of genes coding for proteins involved in the regulation of expression of other genes, hypothesizing that a cascade of effects related to the changed expression of a large battery of different genes might be a response to altered levels of a significantly smaller number of proteins involved in the general control of various stages of gene expression. In fact, recent proteomic [26] and metabolomic [27] studies have confirmed a huge variability of levels of specific polypeptides and lower molecular mass compounds in MPS cells. These changes were likely the effects of the dysregulated expression of many genes, corroborating the idea of the high importance of disturbed gene expression in the MPS pathomechanism. Here, we used fibroblasts derived from patients suffering from 11 types of MPS as model cells, treating them as a group of similar diseases of a common primary cause (GAG storage).

## 2. Materials and Methods

### 2.1. Cell Lines and Culture Conditions

As models of MPS, fibroblasts obtained from patients with different MPS types were used. MPS cell lines were obtained from the NIGMS Human Genetic Cell Repository at the Coriell Institute for Medical Research (which also have all required approval in the light of bio-ethical standards). The following lines of fibroblasts were employed (see ref. [23] for more details): MPS I (catalogue number of the Coriell Institute: GM00798, sex: female, age at the time of sample collection: 1 year, mutated gene: *IDUA*, mutations: p.Trp402X/p.Trp402X; such a genotype, with two non-sense mutations, implicates the severe clinical subtype, MPS I-H, called Hurler syndrome; thus, the obtained results should be considered specific for this subtype which is the most frequent one), MPS II (GM13203, male, 3 years, *IDS*, p.His70ProfsX29/-), MPS IIIA (GM00879, female, 3 years, *SGSH*, p.Glu447Lys/p.Arg245His), MPS IIIB (GM00156, male, 7 years, *NAGLU*, p.Arg626Ter/p.Arg626Ter), MPS IIIC (GM05157, male, 8 years, *HGSNAT*, p.Gly262Arg/pArg509Asp), MPS IIID (GM05093, male, 7 years, *GNS*, p.Arg355Ter/p.Arg355Ter), MPS IVA (GM00593, female, 7 years, *GALNS*, p.Arg386Cys/p.Phe285Ter), MPS IVB (GM03251, female, 4 years, *GLB1*, p.Trp273Leu/p.Trp509Cys), MPS VI (GM03722, female, 3 years, *ARSB*, mutations undetermined—diagnosis made on the basis of drastically decreased enzyme activity), MPS VII (GM00121, male, 3 years, *GUSB*, p.Trp627Cys/p.Arg356X), and MPS IX (GM17494, female, 14 years, *HYAL1*, mutations undetermined—diagnosis made on the basis of drastically decreased enzyme activity). In control experiments, the HDFa line of a fibroblast was used. The cultivation of cells was conducted in the DMEM medium. Supplementation with 10% fetal bovine serum and a standard mixture of antibiotics was used routinely. Genistein was added, when indicated, to 50 μM for 48 h in order to impair GAG synthesis [28]. Cultures were incubated at 37 °C with 5% CO_2_ saturation and 95% humidity.

### 2.2. Transcriptomics

Transcriptomic analyses were performed exactly as described earlier [5]. Briefly, RNA was isolated and purified from 5 × 10^5^ cells withdrawn from each culture (between the 4th and 15th passage). Four biological repeats of each isolation, purification, and further analyses were conducted and the results were used for statistical analyses. mRNA libraries were constructed and used for the preparation of cDNA libraries which were then sequenced using HiSeq4000 (Illumina, San Diego, CA, USA). For bioinformatic analyses, 4 × 10^7^ raw reads from each sample were used which gave at least 12 Gb of raw data per sample. The GRCh38 human reference genome from the Ensembl database was used to map the raw readings. The BioMart interface was employed for the annotation and classification of transcripts. Raw data of the RNA-seq analysis are deposited at NCBI Sequence Read Archive (SRA), under accession no. PRJNA562649.

### 2.3. Western Blotting

Levels of selected proteins were determined employing the WES system (WES—Automated Western Blots with Simple Western; ProteinSimple, San Jose, CA, USA) for automatic Western blotting. The separation of proteins was performed using the 12–230 kDa Separation Module with 8 × 25 capillary cartridges (#SM-W004; ProteinSimple, San Jose, CA, USA). Proteins were detected with the following primary antibodies: EXOSC9 rabbit antibody (#A303-888A, Thermo Fisher Scientific, Waltham, MA, USA), RPL23 rabbit antibody (#A305-010A, Thermo Fisher Scientific), SRSF10 rabbit antibody (#42267S, Cell Signaling Technology, Danvers, MA, USA), NOTCH3 rabbit antibody (#2889S, Cell Signaling Technology). For detection, secondary antirabbit antibodies and an Anti-Rabbit Detection Module (#DM-001, ProteinSimple) were employed. Levels of specific proteins were measured with the WES system, using the Total Protein Detection Module for Chemiluminescence (#DM-TP01, ProteinSimple) as a loading control. 

### 2.4. Statistical Analyses

For transcriptomic studies, statistical significance was determined using the results from 4 biological repeats of each RNA isolation procedure (*n* = 4) in the case of each cell line. A one-way analysis of variance (ANOVA) was used on log_2_(1 + x) values with normal continuous distribution. The Benjamini–Hochberg method was employed to calculate the false discovery rate (FDR). A post hoc Student’s *t*-test with a Bonferroni correction was used for comparisons between two groups. These analyses were conducted with the R software v3.4.3, and the significance was assumed if *p* < 0.1, according to standards used in transcriptomic analyses [17,18,19,20,21,22,23,24,25]. In Western blotting experiments, mean values from 3 biological experiments (*n* = 3) ± standard deviation (SD) were used for statistical analyses, and the significance was assumed if *p* < 0.05. 

## 3. Results

### 3.1. Transcriptomic Analyses

Using the Gene Ontology database (http://geneontology.org/ (accessed on 3 March 2022), we assessed levels of transcripts of genes included in the term “gene expression” (GO:0010467 in the QuickGO; https://www.ebi.ac.uk/QuickGO accessed on 3 March 2022) in MPS cells relative to control fibroblasts (HDFa). We found that the expression of dozens of genes from this category significantly changed (either up- or down-regulated) in the cells derived from patients suffering from all MPS types in comparison to the control cells (Figure 1). Among all tested MPS types, the lowest number of transcripts with altered levels occurred in MPS VII (33 transcripts) and the highest number occurred in MPS VI (99 transcripts). These results indicate that the expression of a significant number of genes coding for proteins involved in activities of other genes is affected by MPS, and this was valid for all tested MPS types.

We hypothesized that the previously described [5] changes in the levels of hundreds of transcripts in MPS cells might arise from an alteration in the expression of genes encoding factors involved in gene expression regulation at various stages. If such a hypothesis is true, then changed amounts of regulatory factors could cause further alterations in activities of many (hundreds) different genes. Our analysis indicated that between 23 (in MPS VI) and 67 (in MPS IVB) transcripts from this sub-category (child term GO:0010468: regulation of gene expression) revealed changed levels in MPS cells relative to controls (Figure 2).

In addition, we analyzed the expression of genes classified in other sub-categories of the term “gene expression” (GO:0010467) (child terms), and found that there were significant alterations in the levels of some transcripts in MPS cells (though the numbers were lower than in the case of GO:0010468: regulation of gene expression) belonging to the following terms: GO:0140053: mitochondrial gene expression, GO:0006396: RNA processing, GO:0010628: positive regulation of gene expression, GO:0010629: negative regulation of gene expression, GO:0097659: nucleic acid-templated transcription, GO:0006412: translation, GO:0006406: mRNA export from the nucleus, and GO:0051604: protein maturation (Figure 3).

We assumed that the most pronounced effects on the general activities of various genes should have specific genes coding for proteins involved in different stages of genetic information expression whose transcripts reveal especially high levels of changes. Therefore, transcripts with levels altered (down- or up-regulated) at least four times (i.e., log_2_FC > 2, where FC means ‘fold change’) in MPS cells relative to the HDFa control were identified among those included in GO:0010467 (“gene expression”) (Table 1). Although the 4-fold change was chosen arbitrarily, on the basis of previous studies one can indicate that this is a change level showing a significant and unambiguous alteration in gene expression which can affect cellular functions [18,19,20,21,22,23,24,25].

Thirty-four transcripts fulfilled such a requirement, and they represented 28 genes (there were two or more kinds of transcripts of some genes) for factors involved in gene expression processes at different stages such as signal transduction (e.g., NOTCH3), transcription initiation (e.g., HOXC9), splicing (e.g., SRSF10), RNA degradation (e.g., EXOSC9), and translation (e.g., RPL23) (Table 1). Interestingly, in all transcripts but one (the *COL4A2* first transcript), the direction of the change (down- or up-regulation) was the same for all MPS types which indicated the general tendency occurring in MPS irrespective of the specific type of the disease (Table 1). On the other hand, there were also clear differences between MPS types, as some genes represented by more than one transcript (*MME*, *RPL10*, *COL4A2*, and *SPOCD1*) revealed various levels of expression dysregulation in fibroblasts derived from patients with specific syndromes (Table 1). This might suggest that the kind(s) of stored GAG(s) might indirectly influence specific mechanisms regulating transcription or post-transcriptional modifications (including RNA degradation).

For further analyses, we chose transcripts of four genes coding for proteins operating at various stages of the process of gene expression. They were characterized as having experienced significant changes in at least six MPS types. These genes were as follows: *EXOSC9*, coding for a component of the exoribonuclease complex which degrades RNA molecules [29], *RPL23*, coding for one of ribosomal subunits [30], *SRSF10*, encoding a protein involved in RNA splicing [31], and *NOTCH3*, encoding a receptor involved in the signal transduction process which regulates the activities of transcription factors [32]. The changes in the levels of these transcripts in fibroblasts representing all MPS types are summarized in Table 2. Despite the fact that the same tendency of changes in the levels of specific transcripts could be observed in all MPS types, the evident differences between the diseases might again suggest an influence of the kind(s) of stored GAG(s) on the gene expression regulation in the tested cells.

### 3.2. Levels of Selected Proteins in MPS Cells

To investigate the expression of the selected genes (*EXOSC9, RPL23, SRSF10, NOTCH3*) in more detail, we measured the levels of their products in MPS cells relative to control cells. Amounts of protein were determined by Western blotting in all MPS types, and the results of these analyses are shown in Figure 4. Interestingly, we observed different levels of compatibility between the results of the transcriptomic analysis of the expression of specific genes, and the levels of protein products of these genes measured with the immunochemical method. Specifically, the levels of the *EXOSC9* transcripts were significantly decreased in MPS I, MPS IIIA, MPS IIIB, MPS IIIC, MPS IIID, MPS IVA, MPS IVB, and MPS IX fibroblasts relative to control cells, and these changes reflected significantly decreased amounts of the EXOSC9 protein in MPS I, MPS IIIB, MPS IIID, and MPS IX (compare Table 2 and Figure 4). This indicated a relatively high compatibility of the results of these two kinds of experiments. However, in the case of the *RPL23* gene and its product, we observed a significant decrease in the corresponding transcript levels in MPS II, MPS IIIC, MPS IIID, MPS IVA, MPS IVB, MPS VI, and MPS VII, whereas the levels of the protein synthesized as a result of the activity of this gene were increased rather than decreased in MPS I, MPS IIIA, MPS IIIB, MPS IIIC, MPS IVA, MPS IVB, MPS VII, and MPS IX. Thus, post-transcriptional regulation appears to be crucial in the control of the expression of *RPL23*. Transcripts of *SRSF10* were down-regulated in MPS I, MPS II, MPS IIIA, MPS IIIB, MPS VI, and MPS IX, while levels of the SRSF10 protein were increased in MPS I and MPS II though they were significantly decreased in other MPS types, indicating a partial compatibility between the two kinds of experiments. Finally, *NOTCH3* transcript levels were increased in all MPS types but MPS II, MPS IVA, and MPS VI, whereas elevated amounts of the NOTCH3 protein were detected only in MPS I and MPS II, in contrast to a significant decrease observed in MPS IIID, MPS IVB, MPS VII, and MPS IX, also pointing to the partial accordance between the two methods of the estimation of gene expression efficiency (Table 2 and Figure 4).

### 3.3. Effects of the Reduction in the GAG Levels in MPS Cells on the Abundance of the Selected Proteins

To test whether the elimination of the GAG storage in MPS cells can correct the levels of the selected proteins in the cells, we used genistein. This isoflavone has been demonstrated previously to significantly reduce the amounts of accumulated GAGs in MPS fibroblasts through the inhibition of their synthesis [28]. Therefore, MPS cells of types in which changed levels of EXOSC9, RPL23, SRSF10, and NOTCH3 were detected (see Figure 4) were treated with 50 μM genistein for 48 h, and the effects on the abundance of these proteins were determined by Western blotting. The results of these experiments are demonstrated in Figure 5 (for measurement of EXOSC9), Figure 6 (for RPL23), Figure 7 (for SRSF10), and Figure 8 (for NOTCH3). 

Levels of EXOSC9 decreased in MPS cells relative to HDFa controls while treatment with genistein, in order to reduce GAG levels, resulted in an improvement of this parameter in MPS I and MPS IIID while it remained at the level similar to (without statistically significant differences) cells untreated with this isoflavone in MPS IIIB and MPS IX (Figure 5). The amount of the RPL23 protein in cells increased in several MPS types but decreased in MPS IX after genistein treatment and were corrected in MPS IIIA, MPS IIIB, MPS IVB, and MPS VII, further worsened in MPS I, and were not affected in MPS IIIC, MPS IVA, and MPS IX (Figure 6). The SRSF10 protein was characterized by elevated levels in MPS I and MPS II, and the amount decreased in all other MPS types. Contrary to other tested proteins, no improvement in the levels of SRSF10 was observed in any MPS types after reducing the GAG concentrations by genistein, with an even worse storage in MPS IIIB, MPS IIIC, MPS IVA, MPS IVB, MPS VII, and MPS IX and no significant changes in MPS I, MPS II, MPS IIIA, and MPS VI (Figure 7). MPS I and MPS II were the only types of the disease in which an increase in the NOTCH3 protein level was observed without genistein while in other types, this protein was more abundant than in control HDFa cells. However, treatment with genistein resulted in the normalization of NOTCH3 amounts in MPS II, MPS IIID, MPS VII, and MPS IX, a worsening in MPS IVB, and no changes in MPS I (Figure 8). These results indicate a high variability in responses to the genistein-mediated reduction in the GAG storage of different MPS types and of various levels of proteins. 

## 4. Discussion

MPS are monogenic diseases caused by mutations in the genes involved in GAG metabolism, and it is commonly accepted that lysosomal GAG storage, resulting from the dysfunction of one of enzymes required for the degradation of these compounds, is the primary metabolic defect responsible for the severe symptoms found in patients [7]. However, recent studies have indicated that secondary and tertiary effects can considerably modulate the course of the disease due to various cellular dysfunctions arising not necessarily directly from the storage, but rather from the disruption of various processes in the cascade of changes in the regulatory mechanisms [17]. In this light, a failure to correct all symptoms in MPS patients treated with HSCT, ERT, SRT, or gene therapy, which theoretically should remove the primary metabolic defect [9,10,11,12,13,14,15,16], might be recognized from another point of view. The discovery of significant changes in the expression of ~300–900 genes in MPS cells [5] indicated that the disrupted control of activities of many genes can contribute considerably to the disease pathomechanism. However, it was unclear how could such huge effects on the expression of hundreds of genes result from GAG storage. 

In this report, we have demonstrated that the activities of many genes coding for proteins involved in different stages of expression of other genes are affected in MPS cells relative to control cells (Figure 1, Figure 2 and Figure 3). Analyses of transcripts with especially high changes (FC > 2) in levels (Table 1) showed that they include those coding for proteins required at various stages of gene expression, such as signal transduction, transcription, splicing, RNA degradation, translation, and others. Therefore, we hypothesized that different amounts of such proteins, controlling the expression of other genes, might strongly contribute to the dysregulation of hundreds of genes in MPS cells (as observed previously [5,18,19,20,21,22,23,24,25,26,27]). For more detailed analyses, the following genes (and corresponding gene products) were chosen: *EXOSC9* (coding for a structural component of the exosome which functions in the RNA degradation process [29]), *RPL23* (coding for a large ribosomal subunit protein L23 [30]), *SRSF10* (coding for a splicing regulatory protein [31]), and *NOTCH3* (coding for a receptor protein involved in cellular signaling [32]). Apart from demonstrating especially high changes in the levels of transcripts of these genes in MPS cells, we also showed significant changes in the amounts of the corresponding proteins in these cells. Interestingly, these changes were not always compatible, i.e., in some cases, the directions of changes (up- or down-regulation) in the levels of the transcripts were opposite to those in the levels of proteins (Table 2 and Figure 4, Figure 5, Figure 6, Figure 7 and Figure 8). Thus, for these genes, the regulation of their expression proceeds at both the transcriptional and pos-transcriptional stages, perhaps also including the control of translation.

Interestingly, it has been demonstrated previously that mutations in genes which were analyzed in more detail in MPS and revealed high levels of changes in their expression (*EXOSC9, RPL23*, *SRSF10*, and *NOTCH3*) result in the development of various diseases whose symptoms resemble those observed in MPS. Namely, *EXOSC9* dysfunction leads to a neurodegenerative disease pontocerebellar hypoplasia type 1b [29], and neurodegeneration occurs in most MPS types [8]. Thus, one might assume that the downregulation of *EXOSC9* expression (Figure 5) can contribute to neurodegenerative processes in MPS patients. Moreover, RLP23 has been reported as a factor involved in various cellular processes, such as cell proliferation, apoptosis, and cell cycle, and all of them were demonstrated to be affected in MPS cells [18,21,33]. Furthermore, the SRSF10 protein is important for neurological processes and responses to viral infections [31], and abnormalities of both these groups of processes are evident in MPS [8,24]. Finally, mutations in the *NOTCH3* gene cause various abnormalities, including arteriopathy and leukoencephalopathy, which occur also (to various degrees) in MPS [8], thus suggesting that the dysregulation of expression of this gene may be connected to MPS symptoms.

From the point of view of the development of effective therapies for MPS, it was important to test if a reduction in GAG storage results in the normalization of expression of the tested genes, measured by the levels of their final products—specific proteins. GAG levels were reduced by using genistein, a compound described previously by different research groups as an effective molecule in impairing the efficiency of GAG synthesis and decreasing the levels in cells [28,34,35,36,37]. Perhaps surprisingly, treatment with genistein corrected the levels of expression of *EXOSC9, RPL23*, *SRSF10*, and *NOTCH3* genes only in some tested MPS lines but not in others (Figure 5, Figure 6, Figure 7 and Figure 8). Therefore, changes in the expression of these genes could be more stable than expected, and they might not necessarily be normalized if the primary metabolic defect is reversed. As a result, under the “therapeutic” conditions, sub-optimal levels of corresponding proteins (*EXOSC9*, *RPL23*, *SRSF10*, and *NOTCH3*) could still affect the expression of other genes, preventing the disappearance of cellular defects, and further correcting tissue and organ dysfunctions. Levels of these proteins in animal MPS models or MPS patients subjected to various therapeutical procedures remain to be determined. However, if this hypothesis is true it can explain, at least partially, a failure to reverse all MPS symptoms by any therapeutical approaches used to date, despite the efficient reduction in GAG storage.

The main limitations of this study were the use of a single cell line from each MPS type and the use of fibroblasts as the only kind of cells. Obviously, to perform a comprehensive study on a given disease, biological material from many patients should be employed, including samples from individuals representing different courses of the disease (mild, intermediate, severe). Moreover, fibroblasts are only representative and are model cells, while MPS affects virtually all kinds of cells. On the other hand, one must note that MPS are rare diseases with a severe course; thus, obtaining biological material from many patients is extremely difficult, if not impossible. For example, to date, only four patients suffering from MPS IX have been reported worldwide [8], making studies on a higher number of patients objectively impossible. Due to the severity of the disease, a lack of cooperativity among the vast majority of patients, and the related ethical aspects (unnecessary medical interventions), obtaining any biological material from MPS patients is difficult and problematic. In this light, we chose to use one cell line per each MPS type; however, our analyses indicated that the directions of changes in levels of transcripts of most genes are the same in various MPS types (Table 1 and Table 2) which indicates that the regulatory mechanisms might be similar in the whole group of the diseases. Moreover, each experiment with RNA isolation was repeated four times (four independent biological repeats) making the results of the performed studies reliable. One should also note that fibroblasts have previously been used as model cells in many studies on MPS which has led to the discovery of the general mechanisms operating in this disease [8]. Therefore, if the conclusions do not concern the specific symptoms of patients but are rather restricted to the molecular mechanisms of the disease, such cells appear to be appropriate research models.

## 5. Conclusions

Changes in the expression of hundreds of genes in MPS cells may result from the dysregulation of activities of a smaller group of genes whose products are involved in the expression of other genes at various stages of this process. Dysregulation of gene expression in MPS cells is not fully normalized by the reduction in GAG levels, pointing to possible reasons for the failure to correct all MPS symptoms in patients subjected to all types of therapies used to date.

## Figures and Tables

**Figure 1 genes-13-00593-f001:**
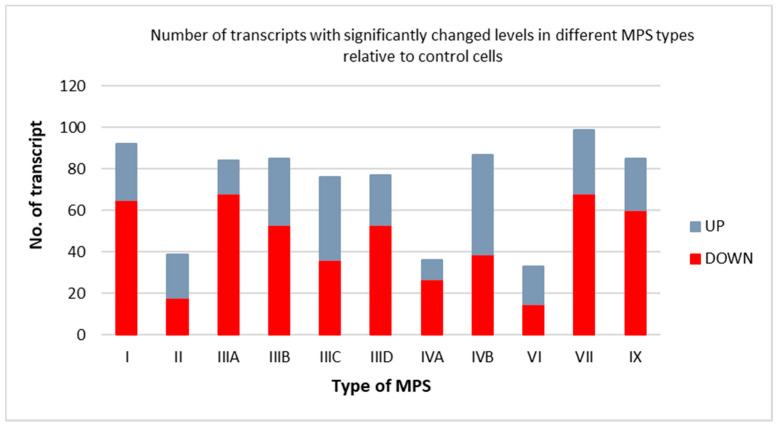
Number of transcripts coding for proteins involved in gene expression (GO:0010467) with changed levels (at FDR < 0.1; *p* < 0.1) in different types of MPS relative to control cells (HDFa). Up-regulation (UP) and down-regulation (DOWN) in MPS relative to HDFa are marked.

**Figure 2 genes-13-00593-f002:**
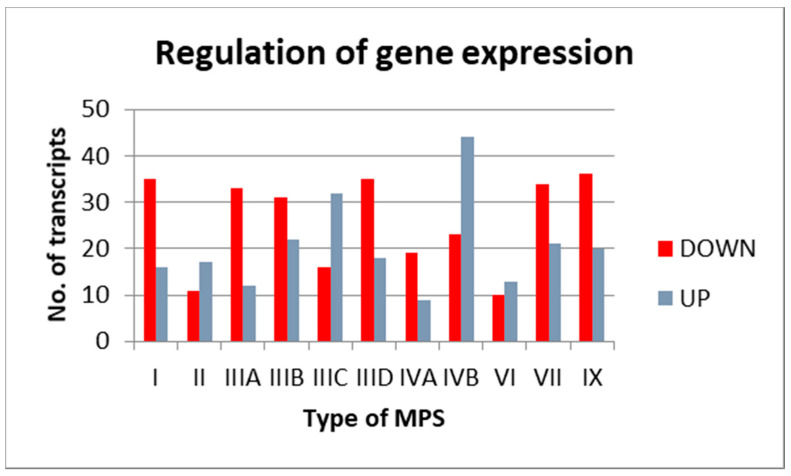
Number of transcripts coding for proteins involved in the regulation of gene expression (GO:0010468) with changed levels (at FDR < 0.1; *p* < 0.1) in different types of MPS relative to control cells (HDFa). Down- and up-regulated transcripts are marked (DOWN and UP, respectively).

**Figure 3 genes-13-00593-f003:**
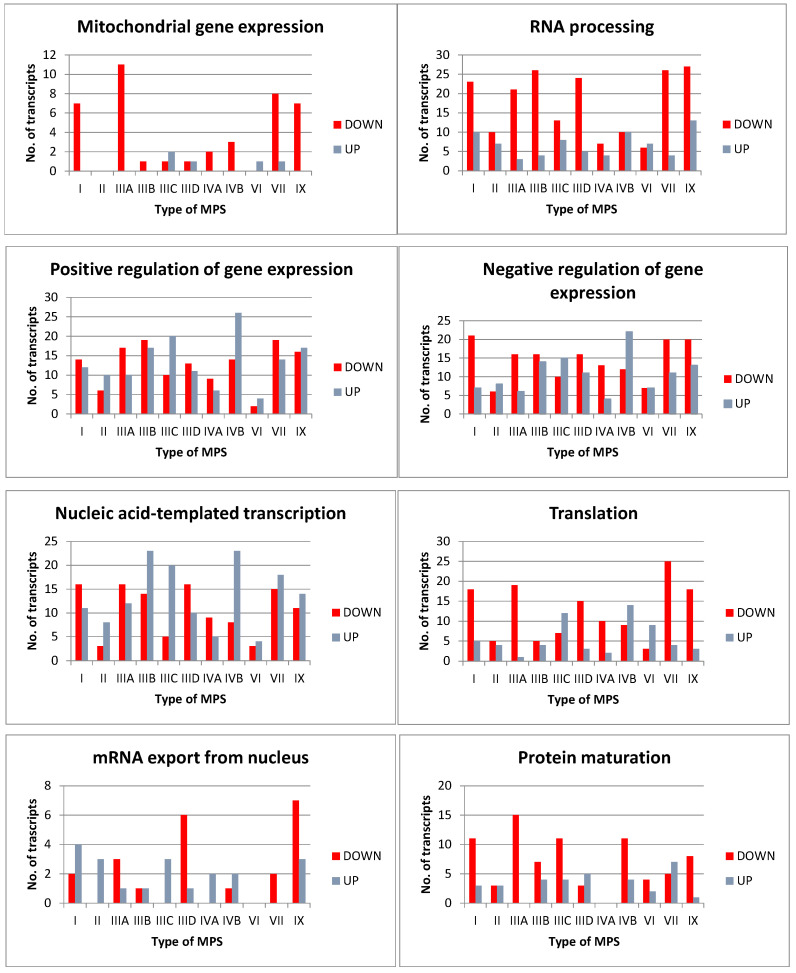
Number of transcripts of genes with changed levels (at FDR < 0.1; *p* < 0.1) in different types of MPS relative to control cells (HDFa) from the following sub-categories of the term “gene expression” (GO:0010467) (child terms): GO:0140053: mitochondrial gene expression, GO:0006396: RNA processing, GO:0010628: positive regulation of gene expression, GO:0010629: negative regulation of gene expression, GO:0097659: nucleic acid-templated transcription, GO:0006412: translation, GO:0006406: mRNA export from the nucleus, and GO:0051604: protein maturation. Down- and up-regulated transcripts are marked (DOWN and UP, respectively).

**Figure 4 genes-13-00593-f004:**
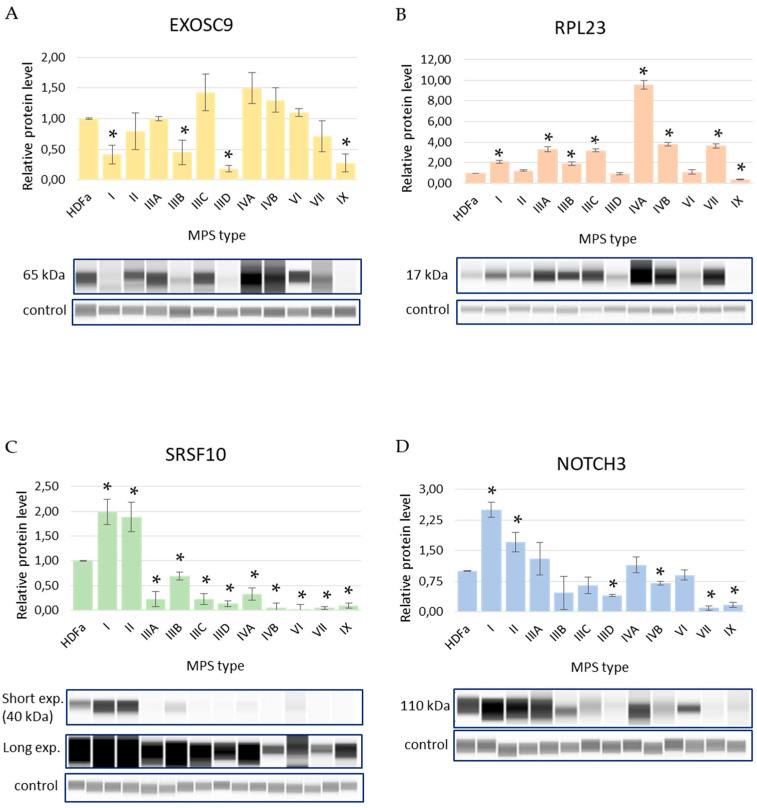
Levels of selected proteins (EXOSC9, panel (**A**); RPL23, panel (**B**); SRSF10, panel (**C**); NOTCH3, panel (**D**)) in fibroblasts of different types of MPS relative to control HDFa cells. The lower parts of each panel represent representative fragments of the blots (‘control’ represents loading control; in panel (**C**), short and long exposures are shown due to the low-level signal under standard conditions), and the upper parts demonstrate the quantification of the results. The results indicated by the columns are the mean values from three independent experiments with error bars representing SD. Statistically significant differences relative to the HDFa control (*p* < 0.05) are indicated by asterisks (*).

**Figure 5 genes-13-00593-f005:**
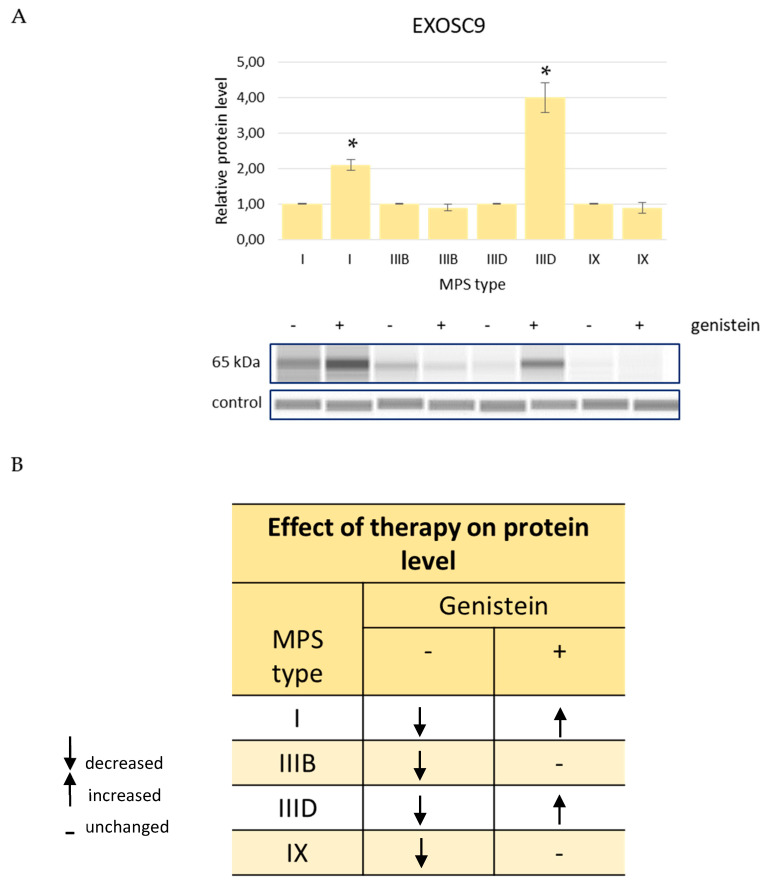
Levels of the EXOSC9 protein in fibroblasts of different types of MPS with (+) and without (−) treatment with 50 μM genistein for 48 h (only MPS types which revealed significantly changed amounts of this protein relative to HDFa control cells (see Figure 4) are shown). Panel (**A**) shows representative blots and the quantification of the results by densitometry, where value 1 corresponds to the level of EXOSC9 determined in MPS cells not treated with genistein. Panel (**B**) shows the scheme of results where the column (−) indicates the levels of EXOSC9 in MPS cells relative to HDFa control while the column (+) indicates the levels of this protein in cells of the corresponding MPS types after 48 h of treatment with 50 μM genistein relative to untreated MPS cells. Up-headed arrows indicate increased levels, down-headed arrows indicate decreased levels, and dashes indicate no significant changes. In panel (**A**), the results indicated by the columns are mean values from three independent experiments with error bars representing SD. Statistically significant differences relative to the untreated MPS cells (*p* < 0.05) are indicated by asterisks (*).

**Figure 6 genes-13-00593-f006:**
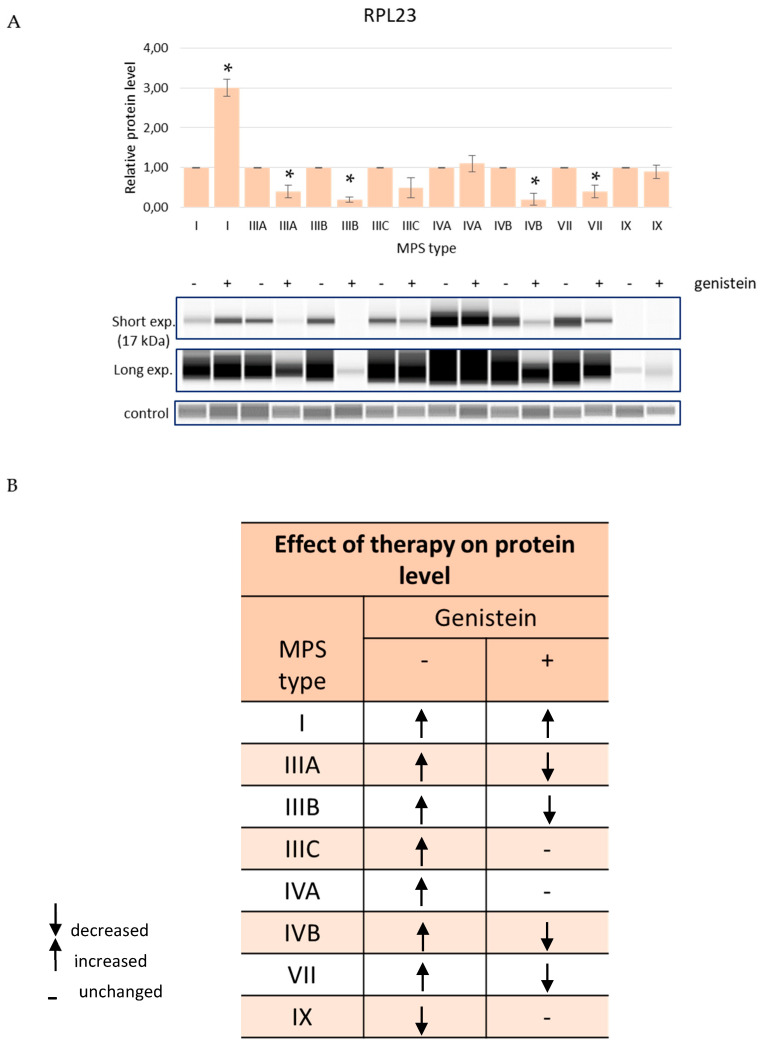
Levels of the RPL23 protein in fibroblasts of different types of MPS with (+) and without (−) treatment with 50 μM genistein for 48 h (only MPS types which revealed significantly changed amounts of this protein relative to HDFa control cells (see Figure 4) are shown). Panel (**A**) shows representative blots and the quantification of the results by densitometry, where value 1 corresponds to the level of RPL23 determined in MPS cells not treated with genistein. Panel (**B**) shows the scheme of results where the column (−) indicates the levels of RPL23 in MPS cells relative to HDFa control while the column (+) indicates the levels of this protein in cells of the corresponding MPS types after 48 h of treatment with 50 μM genistein relative to untreated cells. Up-headed arrows indicate increased levels, down-headed arrows indicate decreased levels, and dashes indicate no significant changes. In panel (**A**), the results indicated by columns are mean values from three independent experiments with error bars representing SD. Statistically significant differences relative to the untreated MPS cells (*p* < 0.05) are indicated by asterisks (*).

**Figure 7 genes-13-00593-f007:**
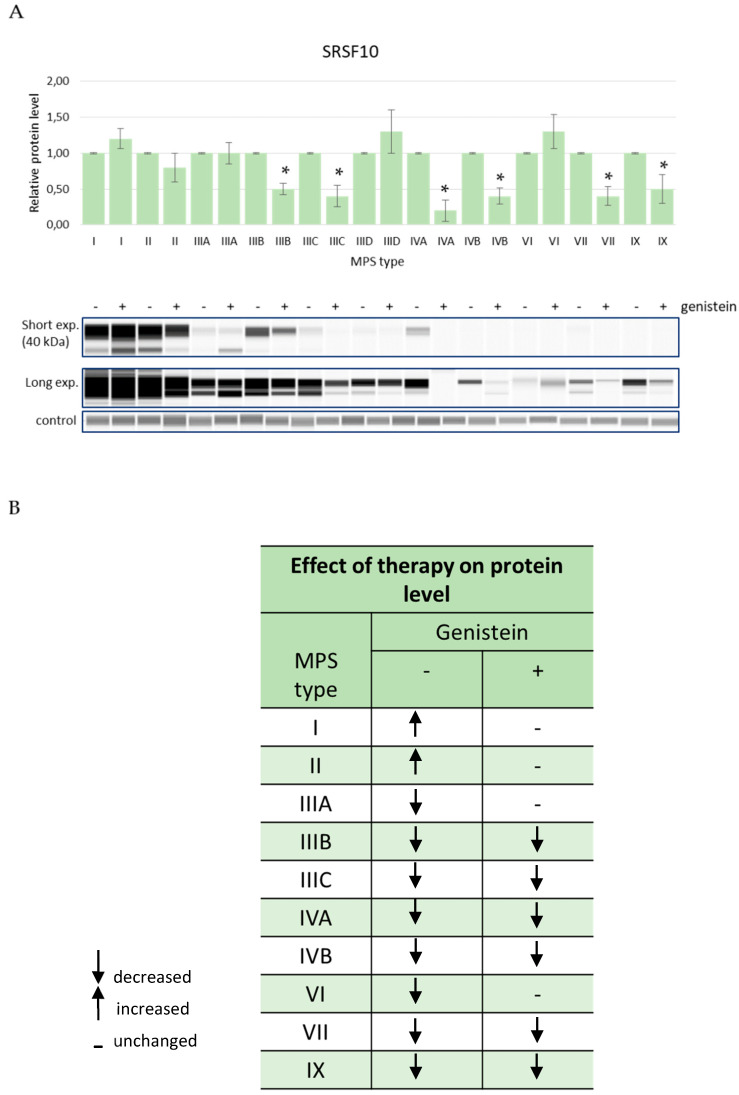
Levels of the SRSF10 protein in fibroblasts of different types of MPS with (+) and without (−) treatment with 50 μM genistein for 48 h (only MPS types which revealed significantly changed amounts of this protein relative to HDFa control cells (see Figure 4) are shown). Panel (**A**) shows representative blots and the quantification of the results by densitometry, where value 1 corresponds to the level of SRSF10 determined in MPS cells not treated with genistein. Panel (**B**) shows the scheme of results where the column (−) indicates the levels of SRSF10 in MPS cells relative to HDFa control while the column (+) indicates the levels of this protein in cells of the corresponding MPS types after 48 h of treatment with 50 μM genistein relative to untreated MPS cells. Up-headed arrows indicate increased levels, down-headed arrows indicate decreased levels, and dashes indicate no significant changes. In panel (**A**), the results indicated by the columns are mean values from three independent experiments with error bars representing SD. Statistically significant differences relative to the untreated MPS cells (*p* < 0.05) are indicated by asterisks (*).

**Figure 8 genes-13-00593-f008:**
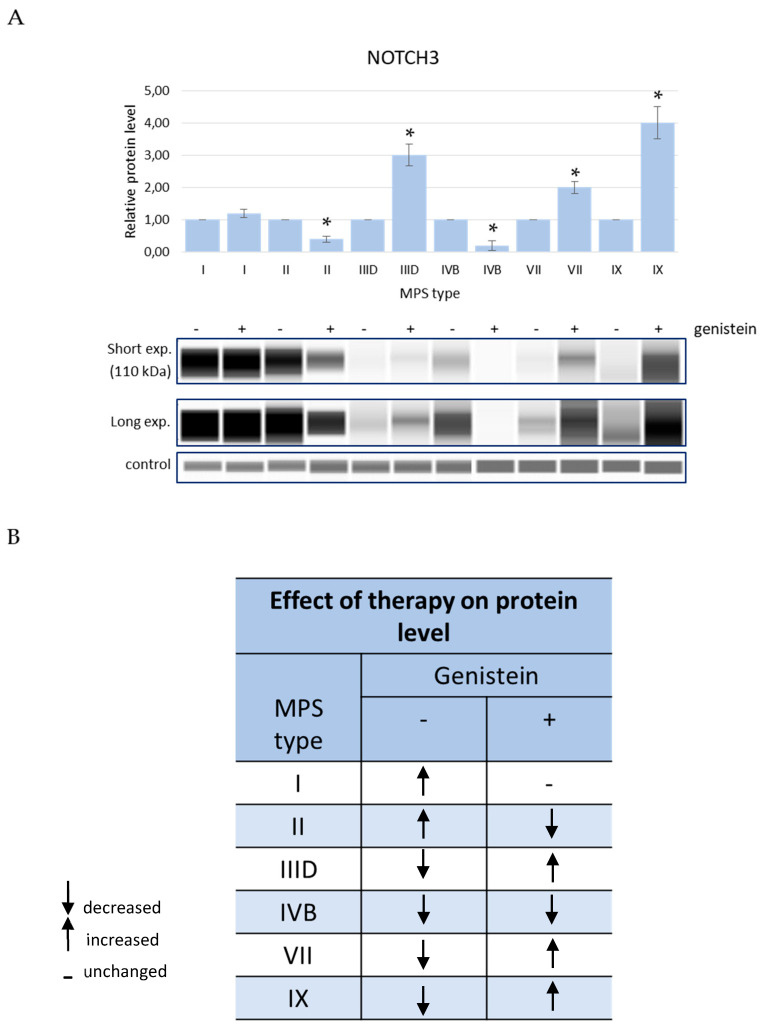
Levels of the NOTCH3 protein in fibroblasts of different types of MPS with (+) and without (−) treatment with 50 μM genistein for 48 h (only MPS types which revealed significantly changed amounts of this protein relative to HDFa control cells (see Figure 4) are shown). Panel (**A**) shows representative blots and the quantification of the results by densitometry, where value 1 corresponds to the level of NOTCH3 determined in MPS cells not treated with genistein. Panel (**B**) shows the scheme of results where the column (−) indicates the levels of NOTCH3 in MPS cells relative to HDFa control while the column (+) indicates the levels of this protein in cells of the corresponding MPS types after 48 h of treatment with 50 μM genistein relative to untreated MPS cells. Up-headed arrows indicate increased levels, down-headed arrows indicate decreased levels, and dashes indicate no significant changes. In panel (**A**), the results indicated by the columns are mean values from three independent experiments with error bars representing SD. Statistically significant differences relative to the untreated MPS cells (*p* < 0.05) are indicated by asterisks (*).

**Table 1 genes-13-00593-t001:** Transcripts in which log_2_FC exceeded 2.0 in any MPS type vs. control cell line.

Transcript ^1^	Transcripts with Especially Changed Levels (log_2_FC > 2) in Different MPS Types Relative to Control Cells ^2^										
	I	II	IIIA	IIIB	IIIC	IIID	IVA	IVB	VI	VII	IX
*AEBP1*				↓							
*AKT1*						↓					
*CDKN1A*	↓						↓		↓		
*DEK*		↓									
*DHCR24*										↓	
*F3*										↓	
*EXOSC9*	↓				↓						
*HNRNPF*				↓							
*MME* (tr. 1)						↓					
*MME* (tr. 2)			↓								
*MME* (tr. 3)			↓								
*MME* (tr. 4)	↓					↓					
*RPL23*		↓			↓	↓	↓	↓	↓	↓	
*NUP88*						↓					
*RPL10* (tr. 1)		↓				↓				↓	
*RPL10* (tr. 2)						↓					
*HOXC9*							↓				
*SNAPC1*											↓
*RPP25*			↓					↓			
*EIF1AX*						↓					
*COL4A2* (tr. 1)			↓	↑							
*COL4A2* (tr. 2)				↑							
*FLNA*					↑						
*COMP*					↑	↑					
*CPE*		↑							↑		
*CSDC2*						↑					
*GATA2*				↑	↑			↑			
*NOTCH3*	↑			↑	↑	↑				↑	↑
*POLDIP3*											↑
*RPLP2*	↑			↑							
*SCG5*										↑	
*SPOCD1* (tr. 1)										↑	
*SPOCD1* (tr. 2)										↑	
*CEBPD*								↑			

^1^ If more than one transcript of a given gene could be identified in the RNA-seq analysis, they were indicated as transcript 1 (tr. 1), transcript 2 (tr. 2), and so on. Raw data for all transcripts are available at NCBI Sequence Read Archive (SRA), under accession no. PRJNA562649. ^2^ Down-regulated transcripts are marked with down-headed arrows (↓), and up-regulated transcripts are marked by up-headed arrows (↑).

**Table 2 genes-13-00593-t002:** Genes coding for proteins involved in gene expression regulation whose transcripts revealed significantly changed levels in at least six MPS types relative to the control cells. The colored boxes indicate statistically significant differences (at FDR < 0.1; *p* < 0.1) between MPS and control cell lines: red boxes indicate down-regulation, and blue boxes indicate up-regulation relative to control.

Transcript	Log_2_FC of Levels of Selected Transcripts in Different MPS Types vs. HDFa Cells										
	I	II	IIIA	IIIB	IIIC	IIID	IVA	IVB	VI	VII	IX
*EXOSC9*	−2.32	−1.18	−1.02	−1.69	−2.33	−1.99	−1.72	−1.10	−1.58	−0.48	−1.60
*RPL23*	−0.08	−2.71	−0.18	−4.15	−4.24	−4.16	−4.09	−3.72	−3.07	−3.85	0.10
*SRSF10*	−1.08	−0.74	−0.69	−0.74	−0.69	−0.66	−0.81	−0.72	−0.56	−0.54	−0.93
*NOTCH3*	2.42	0.97	1.93	3.66	2.56	2.93	2.70	1.97	2.01	3.32	2.70

## Data Availability

The raw RNA-seq results are deposited at NCBI Sequence Read Archive (SRA), under accession no. PRJNA562649.

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
