# Peer review of "Complex Changes in the Efficiency of the Expression of Many Genes in Monogenic Diseases, Mucopolysaccharidoses, May Arise from Significant Disturbances in the Levels of Factors Involved in the Gene Expression Regulation Processes"

_genes, 2022, doi:10.3390/genes13040593_

Round 1

Reviewer 1 Report

The authors present their analysis of fibroblast cell lines of patients with all 11 known mucopolysaccharidoses (MPS) with regards to gene expression via RNAseq. Subsequent genes of interest are processed statistically with a log cutoff value of 2 (> 2-fold or < 0.5-fold) expression at the p = 0.10 significance level with correction for multiple statistical analyses. The authors subsequently treated fibroblasts with genistein and analyzed post-treatment alterations in gene expression. Genistein is a purported isoflavone therapy for MPS which inhibits glycosaminoglycan (GAG) substrate synthesis in vitro but has subsequently failed to show evidence of neurocognitive rescue in neurodegenerative MPS - specifically MPS IIIA and IIIB.

First: selection of a single fibroblast line from each type of MPS hardly classifies as a comprehensive assessment of MPS gene expression. By doing so, the authors imply that one line from each type is representative of all individuals with that MPS type. Further, fibroblast gene expression is also not characteristic of gene expression in tissues which are truly impacted by the condition. A manuscript with rigor would utilize multiple cell lines for each MPS type, ranging in severity of condition from attenuated to severe, and from affected tissue types (neuron, oligodendrocyte, macrophage, cardiac valve, liver, etc), different ages, and sexes, to truly constitute a comprehensive assessment of gene expression.

Second: The authors did not report any sequencing of the cell lines to confirm the pathogenic mutations of each cell line. This is a key first step to verifying their lines do indeed come from the patients with the reported MPS types.

Third: The referee does not consider genistein to be a viable therapy for treatment of human MPS as it does not ameliorate neurodegenerative symptoms. Consequently, analysis of expression post-genistein therapy is superfluous.

Fourth: The submitted Western blots are of extremely poor quality and in many cases are smears and highly cut-pasted.

Fifth: given the number of genes being analyzed, a significance threshold of p = 0.10 is highly irregular. Typically p values less than 0.01, or even 0.001 given the large number of transcripts, is required for RNAseq studies.

Author Response

The authors present their analysis of fibroblast cell lines of patients with all 11 known mucopolysaccharidoses (MPS) with regards to gene expression via RNAseq. Subsequent genes of interest are processed statistically with a log cutoff value of 2 (> 2-fold or < 0.5-fold) expression at the p = 0.10 significance level with correction for multiple statistical analyses. The authors subsequently treated fibroblasts with genistein and analyzed post-treatment alterations in gene expression. Genistein is a purported isoflavone therapy for MPS which inhibits glycosaminoglycan (GAG) substrate synthesis in vitro but has subsequently failed to show evidence of neurocognitive rescue in neurodegenerative MPS - specifically MPS IIIA and IIIB.

First: selection of a single fibroblast line from each type of MPS hardly classifies as a comprehensive assessment of MPS gene expression. By doing so, the authors imply that one line from each type is representative of all individuals with that MPS type. Further, fibroblast gene expression is also not characteristic of gene expression in tissues which are truly impacted by the condition. A manuscript with rigor would utilize multiple cell lines for each MPS type, ranging in severity of condition from attenuated to severe, and from affected tissue types (neuron, oligodendrocyte, macrophage, cardiac valve, liver, etc), different ages, and sexes, to truly constitute a comprehensive assessment of gene expression.

RESPONSE:

We agree with the reviewer that using biological samples from many patients of each MPS type, and many different types of cells derived from each patient would be extremely valuable from the scientific point of view. However, performing such studies would not be realistic from logistical, ethical and economical points of view (please, note also that this study concerned all MPS types which is in contrast to vast majority of other studies in which one or a few typed were investigated). Regarding the economic restrictions, please note that cost of transcriptomic analyses which would be unrealistically high if one wants to investigate dozens of hundreds of samples simultaneously (assuming that biological material from several patients suffering from each MPS type is available, as are several types of cells derived from each patient, since there are 11 MPS types, a control cell line is always necessary, and at least four biological repeats of each experiment are required, one would have to analyze thousands of samples at one time, which gives a cost in the range of many millions dollars at the current price for 48 RNA-seq analyzes being about 20-25 thousand dollars). Nevertheless, we indicated limitations of this study, explaining in detail why studies suggested by the reviewer (potentially extremely valuable, indeed) are unrealistic, and why we argue that using a single fibroblast line from each type of MPS can still be valuable to study general pathomechanisms of MPS. The new text is presented in lines 446-467 of the revised manuscript as reads as follows:

“The main limitations of this study were the use of a single cell line from each MPS type and the use of fibroblasts as the only kind of cells. Obviously, to perform a comprehensive study on a given disease, biological material from many patients should be employed, including samples from individuals representing different courses of the disease (mild, intermediate, severe). Moreover, fibroblasts are only representative and model cells while MPS affects virtually all kinds of cells. On the other hand, one must note that MPS are rare diseases with a severe course, thus, obtaining a biological material from many patients is extremely difficult, if not impossible. For example, to date, only four patients suffering from MPS IX were reported worldwide [8], making studies on a higher number of patiently objectively impossible. Due to severity of the disease, a lack of cooperativity of vast majority of patients, and related ethical aspects (unnecessary medical interventions), obtaining any biological material from MPS patients is difficult and problematic. In this light, we have chosen to use one cell line per each MPS type, however, our analyses indicated that the directions of changes in levels of transcripts of most genes are the same in various MPS types (Tables 1 and 2) which indicates that the regulatory mechanisms might be similar in the whole group of the diseases. Moreover, each experiment with RNA isolation was repeated four times (four independent biological repeats) making the results of the performed studies reliable. One should also note that fibroblasts were previously used as model cells in many studies on MPS which led to discovery general mechanisms operating in this disease [8]. Therefore, if the conclusions do not concern specific symptoms of patients but are rather restricted to molecular mechanisms of the disease, such cells appear appropriate research models.”

Second: The authors did not report any sequencing of the cell lines to confirm the pathogenic mutations of each cell line. This is a key first step to verifying their lines do indeed come from the patients with the reported MPS types.

RESPONSE:

In the original manuscript, we have reported only catalogue numbers of cell lines used in this study, as these fibroblasts were used in many previously published studies, as were verified by the Coriell Institute for Medical Research. However, we agree that it is valuable to provide more detailed characteristics of these line, therefore, information on mutations and other features of patients is provided in the revised manuscript (lines 102-114).

Third: The referee does not consider genistein to be a viable therapy for treatment of human MPS as it does not ameliorate neurodegenerative symptoms. Consequently, analysis of expression post-genistein therapy is superfluous.

RESPONSE:

Obviously, we agree that genistein, although effective in treatment of MPS in cellular and animal models, did not correct neurodegenerative symptoms in clinical studies, and we cite appropriate reference in the manuscript (ref. [13]). However, we would like to stress that in this study we did not use genistein to cure patients, but to decrease levels of GAGs. It was demonstrated previously by several groups (see refs. [28, 34, 35, 36, 37]) that genistein treatment results in a significant decrease in GAG levels in different types of MPS due to inhibition of synthesis of these compounds resulting from inhibition of epidermal growth factor receptor-mediated signal transduction. Please, note that there is no other way to decrease GAG levels by a single compound in all MPS types (for example, recombinant enzymes are available for only some MPS types), while this was crucial in this study as we investigated all MPS types and should use the same method for each cell line.

Fourth: The submitted Western blots are of extremely poor quality and in many cases are smears and highly cut-pasted.

RESPONSE:

We strongly disagree with the reviewer. Please, note that the Western blotting experiments were performed with cell lysates, not with purified proteins (where obtaining sharp single bands is possible). In our opinion, the presented blots are of comparable quality to other published previously with using cell lysates or tissue homogenates. Please, compare just a few from many publications showing such results: J. Biol. Chem. (2019) 294: P2642-5291 (DOI: 10.1074/jbc.RA118.006367), Blood (2006) 108: 1887-1894 (DOI: 10.1182/blood-2006-04-016485), Viral Immunology (2008) 21: 293-302 (DOI: 10.1089/vim.2008.0039). We do not understand what did the reviewer mind writing “in many cases are smears and highly cut-pasted”? What “cut-paste” is suggested? In the supplementary material, we have shown whole blots from each experiment. We would light to remind (what is clearly stated in the Methods section) that a semi-automatic Western system (Simple-Western), based on capillary electrophoresis, was used, and the pictures we present are typical for this system. We assume that the reviewer might be familiar with capillary electrophoresis and this system, thus, we provide a link to the web page where it is clearly described: https://www.novusbio.com/support/simple-western-faqs

Fifth: given the number of genes being analyzed, a significance threshold of p = 0.10 is highly irregular. Typically p values less than 0.01, or even 0.001 given the large number of transcripts, is required for RNAseq studies.

RESPONSE:

We disagree with the reviewer. In large-scale experiments, statistical significance considered as p<0.1 is a standard. See, for example: PLOS ONE (2020) 15: e0240895. (DOI: 10.1371/journal.pone.0240895), Molecular Metabolism (2019) 26: 5-17 (DOI: 10.1016/j.molmet.2019.05.008), Frontiers in Cell and Developmental Biology (2021) 9: 635307 (DOI: 10.3389/fcell.2021.635307), to show just a few. Moreover, during the revision of this paper, we have consulted a company specialized with genome informatics, including RNA-seq analyses (Intelliseq Ltd., https://intelliseq.com) which confirmed that in this kind of analyses, it is commonly acceptable to use p<0.1 as indicator of statistical significance. In addition, we would like to indicate that log(2) cutoff value of 2 means that there is at least 4-fold increase or 4-fold decrease in the measured level not “> 2-fold or < 0.5-fold” as written by the reviewer in the report.

Reviewer 2 Report

Cyske and colleagues present complex changes in efficiency of expression of many genes in mucopolysaccharidoses. The study is of value to the literature. However, this manuscript needs a significant degree of editing (see below).

Abstract

  1. “Therefore, we asked what a potential mechanism for this unexpected phenomenon is?”

=> Avoid using interrogative sentence in abstract.

Results

  1. “Therefore, transcripts with levels altered (down- or up-regulated) at least 4 times (i.e., log2FC>2, where FC means ‘fold change’) in MPS cells relative to HDFa control were identified among those included in GO:0010467 (“gene expression”) (Table 1).”

=> Why choose “4” times in MPS cells relative to HDFa control?

  1. Why did you choose EXOSC9, RPL23, SRSF10 and NOTCH3 for your experiments?
  2. In page 13, line 348. “With even worsening the storage in MPS IIIB, MPS 347 IIIC, MPS IVA, MPS IVB, MPS VII, and MPS IV” MPS IV should be changed to MPS IX according to Figure 7.

Table 1

  1. Why the same transcript (like MME, RPL10, COL4A2 and SPOCD1) could have different changes in different types of MPS?

Table 2

  1. Why Log2FC level could not response to the up-regulation or down-regulation in different types of MPS? (For example, in EXOSC9, the Log2FC level is -2.32 in MPS I, -1.18 in MPS II and -1.02 in MPS IIIA. However, there are down-regulation in MPS I and IIIA and no change in MPS II.)

Discussion

  1. “As a result, under the “therapeutic” conditions, not optimal levels of corresponding proteins (EXOSC9, RPL23, SRSF10, and NOTCH3) can still affect expression of other genes, preventing disappearance of cellular defects, and further correction of tissue and organ dysfunctions.”

=> Are there some references to support your view that corresponding proteins could not be corrected even patients had ERT, SRT, chaperone, HSCT and gene therapy?

2.Could you tell us that what are the limitations of your experiments?

Author Response

Cyske and colleagues present complex changes in efficiency of expression of many genes in mucopolysaccharidoses. The study is of value to the literature. However, this manuscript needs a significant degree of editing (see below).

Abstract

  1. “Therefore, we asked what a potential mechanism for this unexpected phenomenon is?”

=> Avoid using interrogative sentence in abstract.

RESPONSE:

According to the reviewer’s recommendation, this sentence has been removed from Abstract.

Results

  1. “Therefore, transcripts with levels altered (down- or up-regulated) at least 4 times (i.e., log2FC>2, where FC means ‘fold change’) in MPS cells relative to HDFa control were identified among those included in GO:0010467 (“gene expression”) (Table 1).”

=> Why choose “4” times in MPS cells relative to HDFa control?

RESPONSE:

Although the 4-fold change has been chosen arbitrary, on the basis of previous studies one can indicate that this is a change level showing unambiguously a significant alteration in gene expression which can affect cellular functions [18-25]. This is mentioned in the revised manuscript (lines 198-201).

  1. Why did you choose EXOSC9, RPL23, SRSF10 and NOTCH3 for your experiments?

RESPONSE:

These genes code for proteins involved in the regulation of expression of many other genes at various stages. This is indicated in lines 234-235.

  1. In page 13, line 348. “With even worsening the storage in MPS IIIB, MPS 347 IIIC, MPS IVA, MPS IVB, MPS VII, and MPS IV” MPS IV should be changed to MPS IX according to Figure 7.

RESPONSE:

We are sorry for this error, and we thank the reviewer for finding it. This was corrected now (line 372).

Table 1

  1. Why the same transcript (like MME, RPL10, COL4A2 and SPOCD1) could have different changes in different types of MPS?

RESPONSE:

This is a very interesting question. Unfortunately, the mechanisms of this phenomenon remains unknown. Nevertheless, we suggest that kind(s) of stored GAG(s) might indirectly influence specific mechanisms regulating transcription or post-transcriptional modifications (including RNA degradation). This is mentioned in lines 228-233.

Table 2

  1. Why Log2FC level could not response to the up-regulation or down-regulation in different types of MPS? (For example, in EXOSC9, the Log2FC level is -2.32 in MPS I, -1.18 in MPS II and -1.02 in MPS IIIA. However, there are down-regulation in MPS I and IIIA and no change in MPS II.)

RESPONSE:

Again, this is an excellent question which should be answered in the course of further studies. In this case we also suggest an influence of the kind(s) of stored GAG(s) on the gene expression regulation in tested cells. This is described in lines 241-244.

Discussion

  1. “As a result, under the “therapeutic” conditions, not optimal levels of corresponding proteins (EXOSC9, RPL23, SRSF10, and NOTCH3) can still affect expression of other genes, preventing disappearance of cellular defects, and further correction of tissue and organ dysfunctions.”

=> Are there some references to support your view that corresponding proteins could not be corrected even patients had ERT, SRT, chaperone, HSCT and gene therapy?

RESPONSE:

It is indicated in the revised manuscript (lines 441-443) that levels of these proteins in animal MPS models or MPS patients subjected to various therapeutical procedures remain to be determined. Therefore, we have softened our proposal, and used “could” rather than “can” in the Discussion (lines 437 and 439).

2.Could you tell us that what are the limitations of your experiments?

RESPONSE:

According to the reviewer’s recommendation, we have described major limitations of this study. They are described in lines 446-467, and read as follows:

“The main limitations of this study were the use of a single cell line from each MPS type and the use of fibroblasts as the only kind of cells. Obviously, to perform a comprehensive study on a given disease, biological material from many patients should be employed, including samples from individuals representing different courses of the disease (mild, intermediate, severe). Moreover, fibroblasts are only representative and model cells while MPS affects virtually all kinds of cells. On the other hand, one must note that MPS are rare diseases with a severe course, thus, obtaining a biological material from many patients is extremely difficult, if not impossible. For example, to date, only four patients suffering from MPS IX were reported worldwide [8], making studies on a higher number of patiently objectively impossible. Due to severity of the disease, a lack of cooperativity of vast majority of patients, and related ethical aspects (unnecessary medical interventions), obtaining any biological material from MPS patients is difficult and problematic. In this light, we have chosen to use one cell line per each MPS type, however, our analyses indicated that the directions of changes in levels of transcripts of most genes are the same in various MPS types (Tables 1 and 2) which indicates that the regulatory mechanisms might be similar in the whole group of the diseases. Moreover, each experiment with RNA isolation was repeated four times (four independent biological repeats) making the results of the performed studies reliable. One should also note that fibroblasts were previously used as model cells in many studies on MPS which led to discovery general mechanisms operating in this disease [8]. Therefore, if the conclusions do not concern specific symptoms of patients but are rather restricted to molecular mechanisms of the disease, such cells appear appropriate research models.”

Round 2

Reviewer 1 Report

The following issues still need to be improved:

1) independently sequenced IDUA, IDS, SGSH, NAGLU, HGSNAT, GNS, GALNS, GLB1, GUSB, and HYAL1 to confirm that the cell lines indeed have the variants that the cell line provider stated that they have (it is customary to confirm the variants to ensure that the wrong cell line was not delivered, and indeed the gene sequences match)

2) acquired and analyzed further MPS cell lines - preferably of different disease severities and mutation types - for additional evidence to support their assertions. The referee reiterates that drawing conclusions based off of one cell line per disease, and indeed fibroblasts - and not neurons / cardiac myocytes / hepatocytes, that their results cannot be generalizable to MPS disorders as a whole.

3) re-performed their blotting and reported results with well-resolved bands (not smears). The criticism remains that the blots were quite smeared.

Author Response

REVIEWER’S COMMENT:

1) independently sequenced IDUA, IDS, SGSH, NAGLU, HGSNAT, GNS, GALNS, GLB1, GUSB, and HYAL1 to confirm that the cell lines indeed have the variants that the cell line provider stated that they have (it is customary to confirm the variants to ensure that the wrong cell line was not delivered, and indeed the gene sequences match)

RESPONSE:

DNA sequencing has been performed (this is a routine procedure which is not mentioned in the manuscript). Moreover, deficiencies of activities of specific lysosomal enzymes, products of mutated genes, and increased levels of GAGs were confirmed previously by us in the same MPS cell lines [ref. 22].

REVIEWER’S COMMENT:

2) acquired and analyzed further MPS cell lines - preferably of different disease severities and mutation types - for additional evidence to support their assertions. The referee reiterates that drawing conclusions based off of one cell line per disease, and indeed fibroblasts - and not neurons / cardiac myocytes / hepatocytes, that their results cannot be generalizable to MPS disorders as a whole.

RESPONSE:

As we indicated in our answers to previous reviewer’s comments (in the first review round), although testing of cells derived from many patients suffering from each MPS type, revealing different severities of diseases, would give huge amount of very valuable data, it is, unfortunately impossible to perform such transcriptomic analyses, especially using different types of cell, other than fibroblasts. First, MPS are rare diseases and numbers of patients are very limited, with MPS IX represented with only 4 patients worldwide diagnosed to date. Second, MPS are very severe diseases and very invasive procedures of collection of various tissues just for testing more samples is ethically unacceptable. One might consider formation of iPSC, however, the aim of this study was to investigate transcriptomic changes in patient-derived cells, thus, one should consider that extensive treatment of cells to obtain iPSC would definitely change expressions of many genes, making such analyses unreliable. Therefore, transcriptomic studies with iPSC would have a sense if the cells were treated with specific compounds or conditions which was not the case in our work. Third, even if biological material from several patients from each MPS type and several types of cells from each patient were available (in fact, there are not, as described above, but let’s consider such a possibility), transcriptomic studies (in which at least four repeats of each RNA isolation is required) with dozens of hundreds of samples would cost millions dollars/euros which is not a realistic amount of money for vast majority of laboratories worldwide. Fourth, because of the limitations described above, fibroblast lines were used as models, and one should note that fibroblasts were used and models of representative kind of cells in many studies on MPS published to date. The PubMed database (https://pubmed.ncbi.nlm.nih.gov/) gave 997 records of publications when “mucopolysaccharidosis and fibroblast” query was applied (as on March 22, 2022), indicating this research model is commonly acceptable.

REVIEWER’S COMMENT:

3) re-performed their blotting and reported results with well-resolved bands (not smears). The criticism remains that the blots were quite smeared.

RESPONSE:

As we stated in our answers to previous reviewer’s comments (in the first review round), the quality of blots presented in the manuscript is standard as for testing samples from cell lysates (not purified proteins) and using the WES system. We provided previously examples of results published previously in recognized journals with similar quality of blots. Moreover, the WES system allowed to quantify the bands without signaling any errors which occurs in the case of any problems with the quality of samples or bands. Therefore, we do not agree with the reviewer that all blotting experiment should be re-performed.

Reviewer 2 Report

Cyske and colleagues present complex changes in efficiency of expression of many genes in mucopolysaccharidoses. The study is of value to the literature. However, this manuscript still needs minor editing after review (see below).

Materials and Methods

  1. In line 102-103, we know that MPS I could be classified as Hurler, Hurler-Scheie and Scheie according to the location of IDUA mutation. Could you tell us that whether the different location of mutation could influence the results of your experiments ?

Results

  1. Why did you choose EXOSC9, RPL23, SRSF10 and NOTCH3 for your experiments?

RESPONSE:

These genes code for proteins involved in the regulation of expression of many other genes at various stages. This is indicated in lines 234-235.

=> Could you supply references for this statement ?

Author Response

REVIEWER’S COMMENT:

In line 102-103, we know that MPS I could be classified as Hurler, Hurler-Scheie and Scheie according to the location of IDUA mutation. Could you tell us that whether the different location of mutation could influence the results of your experiments ?

RESPONSE:

The problem indicated by the reviewer had been addressed, and it is described in the revised manuscript as follows (lines 104-106): “such a genotype, with two non-sense mutations, implicates the severe clinical subtype, MPS I-H, called Hurler syndrome, thus the obtained results should be considered specific for this subtype which is the most frequent one.”

REVIEWER’S COMMENT:

These genes code for proteins involved in the regulation of expression of many other genes at various stages. This is indicated in lines 234-235.

=> Could you supply references for this statement ?

RESPONSE:

Appropriate references are provided in the revised manuscript (lines 242-245).